# Weather-Based Predictive Modeling of *Cercospora beticola* Infection Events in Sugar Beet in Belgium

**DOI:** 10.3390/jof7090777

**Published:** 2021-09-18

**Authors:** Moussa El Jarroudi, Fadia Chairi, Louis Kouadio, Kathleen Antoons, Abdoul-Hamid Mohamed Sallah, Xavier Fettweis

**Affiliations:** 1Water, Environment and Development Unit, Department of Environmental Sciences and Management, UR SPHERES, University of Liège, 6700 Arlon, Belgium; fadia.chairi@hotmail.fr (F.C.); ahsallah@uliege.be (A.-H.M.S.); 2Centre for Applied Climate Sciences, University of Southern Queensland, Toowoomba, QLD 4350, Australia; louis.kouadio@usq.edu.au; 3Institut Royal Belge pour l’Amélioration de la Betterave, 3300 Tirlemont, Belgium; k.antoons@irbab.be; 4Laboratory of Climatology, Department of Geography, UR SPHERES, University of Liège, 4000 Liège, Belgium; xavier.fettweis@uliege.be

**Keywords:** *Cercospora beticola*, fungal foliar disease, plant disease risk, integrated plant disease management

## Abstract

Cercospora leaf spot (CLS; caused by *Cercospora beticola* Sacc.) is the most widespread and damaging foliar disease of sugar beet. Early assessments of CLS risk are thus pivotal to the success of disease management and farm profitability. In this study, we propose a weather-based modelling approach for predicting infection by *C. beticola* in sugar beet fields in Belgium. Based on reported weather conditions favoring CLS epidemics and the climate patterns across Belgian sugar beet-growing regions during the critical infection period (June to August), optimum weather conditions conducive to CLS were first identified. Subsequently, 14 models differing according to the combined thresholds of air temperature (T), relative humidity (RH), and rainfall (R) being met simultaneously over uninterrupted hours were evaluated using data collected during the 2018 to 2020 cropping seasons at 13 different sites. Individual model performance was based on the probability of detection (POD), the critical success index (CSI), and the false alarm ratio (FAR). Three models (i.e., M1, M2 and M3) were outstanding in the testing phase of all models. They exhibited similar performance in predicting CLS infection events at the study sites in the independent validation phase; in most cases, the POD, CSI, and FAR values were ≥84%, ≥78%, and ≤15%, respectively. Thus, a combination of uninterrupted rainy conditions during the four hours preceding a likely start of an infection event, RH > 90% during the first four hours and RH > 60% during the following 9 h, daytime T > 16 °C and nighttime T > 10 °C, were the most conducive to CLS development. Integrating such weather-based models within a decision support tool determining fungicide spray application can be a sound basis to protect sugar beet plants against *C. beticola*, while ensuring fungicides are applied only when needed throughout the season.

## 1. Introduction

Cercospora leaf spot (CLS), caused by the fungus *Cercospora beticola* Sacc., is the most widespread and destructive foliar disease of sugar beet (*Beta vulgaris* L.) [1,2,3,4,5,6]. Under favorable environmental conditions, unprotected susceptible cultivars may suffer substantial yield losses up to 40%, and reduction in recoverable sugar yield and sucrose concentration of up to 50% [7,8,9,10]. The genetics and biology of *C. beticola*, as well as the epidemiology of CLS, have been extensively documented (e.g., [2,3,5,11,12,13]). When weather conditions are favorable, *C. beticola* can complete several asexual cycles within a single cropping season under favorable weather conditions and can survive between growing seasons on infected plant residues, primarily as overwintering conidia-producing hyphal structures (pseudostromata) [12,13,14]. Other potential sources of primary inocula include windborne conidia, infested seed or beet roots, dispersal of *C. beticola* through tools and machinery, and stromata from other host plants [4,5,13,15,16]. Optimum conditions for epidemics of CLS include temperatures ranging between 15 °C and 35 °C, leaf wetness, and extended periods of high relative humidity; no sporulation occurs at temperatures below 10 °C or greater than 38 °C [3,17,18,19,20]. Preventive and prudent cultural practices, including rotation with non-host crops, growing disease-resistant cultivars, and application of fungicides with various modes of action, are widely used to reduce inoculum levels in infested residue levels and manage CLS disease.

The European Union (EU) is the world’s leading producer of sugar beet, with approximately 50% of global production [21]. The major production areas are in northern France, Germany, the Netherlands, Belgium, and Poland, where the climate is more suitable [21]. In these countries, sugar beet is commonly cultivated as a spring crop, in rotation with other crops, generally winter wheat or winter barley, and at varying cropping intervals, e.g., 2 to 3 years or more, depending on the country [10,22]. With the end of the EU’s sugar quota system, effective since the end of the 2016/2017 marketing year, there are opportunities for EU sugar beet growers to maximize production to satisfy potential markets both within the EU and elsewhere [23,24]. Efficient management of the production risks, namely those related to pests and diseases, is therefore crucial to ensure the competitiveness, profitability, and sustainability of sugar beet growers under increasingly variable environmental conditions.

To ensure timely, environmentally sound, and efficacious fungicide application while increasing the likelihood of improved beet root yields and sucrose concentration, decision-making tools or systems are used. The core of these systems relies on epidemiological models that aim at a reasonably sound prediction of CLS onset and disease progression. Various models for predicting occurrence of CLS and simulating the progress of CLS severity have been developed [8,25,26,27,28]. Wolf and Verreet [8] proposed an integrated management system for the control of CLS and powdery mildew (caused by *Erysiphe polygoni*) in sugar beet, which relies on the ability to accurately diagnose the beginning of epidemics. In that system, the onset of the CLS epidemic was defined as the point when 50% of the beet plants were infected (disease severity = 0.01%) [8]. Shane and Teng [25] developed an infection prediction model based on the percent disease severity and daily values of infection by *C. beticola;* the daily infection values being calculated from the number of hours per day (midnight to midnight) with relative humidity ≥ 85% and the average temperature during those hours over the previous two days. The model has been widely implemented and integrated with other control methods in the U.S.A. since the 1990s as a guide for fungicide application [18,29]. Likewise, to help manage CLS epidemics in German sugar beet-growing regions, Racca and Jörg [28] proposed the CERCBET 3 model, a modified version of the Rossi and Battilani [26] model. Inputs for the CERCBET 3 model include meteorological parameters (temperature, relative humidity, leaf wetness, and vapor pressure deficit), agronomic field characteristics (e.g., cultivar susceptibility to CLS), and disease incidence. The model is available to German sugar beet growers through a dedicated web-based information system on integrated crop production in Germany (https://www.isip.de/isip/servlet/isip-de (accessed on 16 July 2021)).

To the best of the authors’ knowledge, none of the systems or models thus far developed have been operationally used to control CLS epidemics in Belgium. The main objective of this study was to evaluate the interest of a weather-based modeling approach for predicting infection events of *C. beticola* in Belgian sugar beet-producing regions to improve the management of CLS epidemics throughout the cropping season in real time. Indeed, through reliably and accurately predicting infection events of *C. beticola* based on forecasted weather, fungicide application and timing at disease onset or prior to the development of symptoms according to the local environmental conditions can be improved, thereby avoiding unnecessary applications while efficaciously protecting the plants against the pathogen.

## 2. Materials and Methods

### 2.1. Study Sites and Disease Monitoring

Field experiments were established at 13 sites during the 2018 to 2020 cropping seasons across the sugar beet belt in the Walloon Region of Belgium (Figure 1). In Belgium, sugar beet is typically sown in March–April, with harvest occurring during September–November. The study sites were planted with several sugar beet cultivars showing a range of susceptibility to CLS (Table 1). Trials were planted in a randomized block design with four replicates (one replicate plot size = 5.25 m × 2.70 m). Prior to sowing, soil analyses were carried to determine actual soil nitrogen content and the rate to be applied at the start of the season. During the study period, nitrogen was applied at variable rate according to the site and year, with rates ranging between 50 and 110 kg N ha^−1^. No fungicide was applied to plots monitored during the study. Sowing and harvest methods, as well as crop practices, were typical of sugar beet production in Belgium. The experiments are part of a large trial of the Institut Royal Belge pour l’Amélioration de la Betterave (IRBAB) aimed at improving integrated management of sugar beet foliar diseases.

Visual assessments of CLS of sugar beet were made on a weekly basis between June and mid-September. Symptoms of CLS were identified on the basis of typical circular lesions scattered on the upper leaf surface, exhibiting a tan to grey color in the center and often delimited by tan-brown to reddish-purple rings [12]. Disease incidence (proportion of leaves exhibiting at least one CLS lesion) and disease severity (percent leaf area diseased) were assessed on 25 leaves per replicate (that is, 100 leaves total) that were randomly selected from the central crown of a single plant; very young and old senescent leaves were discarded. Disease severity was assessed using a modified Horsfall–Barratt scale [31], in which an index of 9 corresponds to 0% diseased leaf area, and an index of 1 corresponds to 100% diseased leaf area [32].

### 2.2. Weather Data

Hourly weather data [air temperature (T), relative humidity (RH), and rainfall (R)] from January 2018 to December 2020 were computed using the regional climate model Modèle Atmosphérique Régional (MAR) version 3.9 [33,34]. The MAR was run at a horizontal resolution of 5-km over a domain covering the whole Belgium, with the first 24 h from the daily forecast run (00h GMT) of the global weather forecast model Global Forecast System (GFS) being used as the lateral boundary conditions of MAR. At the beginning of each day, MAR restarts from its previous state without any reinitialization of its atmosphere or its soil/surface. The 6-hourly outputs from GFS were used here to drive the MAR-based time series instead of climate reanalysis (e.g., ERA5) to be in similar conditions if MAR would be used in real-time operational mode. For each of the study sites, time series of weather data were extracted from the nearest grid point of the model integration domain.

### 2.3. Predictive Model Development

A schematic flowchart describing the steps for developing the weather-based model is presented (Figure 2). The modeling approach follows a conceptual approach proposed previously [25,35]. Details of the modeling approach are described in the following sections.

#### 2.3.1. Determination of Weather Conditions Conducive to Infection by *Cercospora beticola*

A detailed analysis using hourly weather data was performed to characterize the optimum classes of combined weather variables conducive to CLS. Based on the climate patterns across Belgian sugar beet-growing regions during the most critical infection period of sugar beet by *C. beticola* (June to August), various combinations of weather variables (R, RH, and T) (Table 2) were evaluated through a frequency analysis over each 6-day period during June–August for each study year. Our analyses focused on the most critical infection period because any infection events of sugar beet by *C. beticola* and subsequent development of CLS could adversely affect the final beet yield and recoverable sugar yields, and the sucrose concentration.

We further analyzed the proportions of hours with dominant classes of RH and T, associated with rainy conditions (R ≥ 0.1 mm/h), to determine those favorable weather conditions conducive to CLS at the study sites. The definition of intervals of weather variables in this step was based on previously reported favorable weather conditions for CLS [2,5,15,18,36].

#### 2.3.2. Determination of Daily Infections of Sugar Beet by *C. beticola* and Calculation of Latency Periods

A given weather-based model consisted of optimum RH and T being met over uninterrupted hours (i.e., 4 h) of R and RH. In this study, an infection is deemed to have occurred when a given combination of optimum weather conditions was met. The latency period was calculated as follows [37,38]:(1)$$\frac{1}{P}=0.00442\times T-0.0238$$
where *P* is the latency period (days), and *T* is the average daily temperature (°C).

CLS symptoms on leaves may appear within 5–11 days after infection, depending on weather conditions; lesions are first visible on the older leaves and later on the younger ones [14]. Starting from each day of infection, the latency period was calculated at a daily time step, as well as its inverse. The day when the sum of the inverses reaches 1 (that is, 100% of latency achieved) corresponds to the date when CLS symptoms become visible.

#### 2.3.3. Model Testing and Validation

Each weather-based model related to each of the combinations of optimum weather variables. All the defined models were evaluated during the testing step; for the validation step, only the top three best performing models (i.e., having good statistical performance indicators) were considered.

Given the change in experimental site each year due to crop rotation, and to increase confidence in the robustness of the models assessed, a principal component analysis (PCA) was performed using T and RH for each site to group sites with similar conditions to allow for the selection of sites to be considered for the testing and validation phases. Values for T and RH were averaged over the daytime (7.00 a.m. to 8.59 p.m.) and nighttime (9.00 p.m. to 6.59 a.m. the following day) periods for each day. The PCA was performed using the package ‘*factoextra*’ [39] in R [40].

The first two axes of the PCA biplot, summarizing the relationships between climate variables and sites, explained 93.9% of total variation (Figure 3). Two main groups of sites were found: the first group included Avernas 2018, Braffe 2018, Briffoeil 2019, Herstappe 2019, Jandrain 2019, and Meux 2019, which were characterized by relatively high RH (≥70%) and high T (≥18 °C) during the daytime and nighttime; the remainder of the sites (Franc-Warret 2018, Perwez 2018, Villers-le-Peuplier 2018, Avernas 2020, Briffoeil 2020, Wagnelée 2020, Rutten 2020) formed the second group which was characterized by relatively low temperature (≤18 °C) during daytime (Figure 3).

Disease incidence data were partitioned into testing (61%, *n* = 8 sites) and validation (39%, *n* = 5 sites) sets, and were used in an independent manner for evaluation/selection and validation purposes, respectively (Table 3). All sites were randomly chosen from the two groups observed through the PCA.

### 2.4. Model Performance Evaluation

Three statistical scores derived from a contingency table analysis were used to assess the performance of the different models [41,42]. These scores were the probability of detection (POD), the false alarm ratio (FAR), and the critical success index (CSI). They were calculated as follows:(2)$$\mathrm{POD}=100\times \frac{\mathrm{SO}}{\mathrm{SO}+\mathrm{NSO}}$$
(3)$$\mathrm{FAR}=100\times \frac{\mathrm{SNO}}{\mathrm{SO}+\mathrm{SNO}}$$
(4)$$\mathrm{CSI}=100\times \frac{\mathrm{SO}}{\mathrm{SO}+\mathrm{SNO}+\mathrm{NSO}}$$
where *SO*, *SNO*, and *NSO* refer to infections simulated and observed, infections simulated but not observed, and infections observed but not simulated, respectively.

POD corresponds to the probability of correctly forecasting the observed event and varies between 0 and 100, with 100 being a perfect score. FAR is the number of times an event is forecast but is not observed, divided by the total number of forecasts of that event. A perfect FAR score is 0. CSI takes into account both false alarms and missed events. It varies between 0 and 100, with 100 being a perfect CSI score.

An analysis of variance (ANOVA) using SPSS 21 (IBM SPSS Statistics 21, Inc., Chicago, IL, USA) was performed to assess the influence of year, site, and sugar beet cultivars (considered as independent variables) on incidence of CLS (dependent variable). A Tukey’s HSD post hoc means separation test (α = 0.05) was used to compare the means. All other statistical analyses and graphical representations were conducted using R (v4.0.0; [40]) and SigmaPlot (v14; Systat Software Inc., San Jose, CA, USA). ArcGis [43] was used for mapping purpose.

## 3. Results

### 3.1. Incidence of Cercospora Leaf Spot during the Study Period

Cercospora leaf spot was generally first observed during the first weeks of July at most of the study sites, regardless of the cropping season (Figure 4, Figure 5 and Figure 6). The disease incidence gradually increased as the season progressed. Highest incidences of CLS were recorded during the 2018 cropping season, with incidence levels ≥75% in September at all sites (Figure 4). Comparatively, the disease incidence was most often less than 50% during the 2020 cropping season (Figure 6). Over the three cropping seasons, the highest incidences of CLS were recorded at Braffe and Perwez (2018), Herstappe (2019), and Briffoeil (2020) (Figure 4, Figure 5 and Figure 6). No statistical difference (*p* > 0.05) was found between cultivars for disease incidence. Nonetheless, the cultivars Acacia and Bayamo had numerically highest incidence during the study period (Figure 4 and Figure 6).

### 3.2. Weather Conditions during the Critical Infection Period of Sugar Beet by Cercospora beticola

To reduce redundancy, only the distribution of weekly total hours of defined ranges of T, RH, and R, during the months of June, July, and August 2018, is presented (Figure 7). Corresponding distributions for the 2019 and 2020 cropping seasons are provided in Appendix A. Over the study period, daytimes were most often warm during the June–August period of the 2018 to 2020 cropping seasons; the class of dominant daytime temperature (DT) at the majority of the study sites was 16–20 °C (Figure 7, Appendix A). Patterns of nighttime temperatures (NT) were similar in 2018 and 2020 at all the study sites, with NT ≥ 15 °C particularly during the months of July and August (Figure 7 and Appendix A). In 2019, nighttime conditions similar to those of 2018 and 2020 were observed at Briffoeil and Meux only; at the other two sites (Herstappe and Jandrain), nighttime was relatively cool (NT < 15 °C) (Appendix A). Humidity conditions varied according to the year and site, with 2018 having more hours with RH ≥ 80% during the first two weeks of June at most of the sites (the only exception was at Braffe) (Figure 7). Hours with no rain were dominant during the 3-month period in 2018, 2019, and 2020 at all the study sites, with more than 100 h on average without rain (Figure 7, Appendix A). In 2018, there were no rainy hours at any site during the fourth week of July (Figure 7).

### 3.3. Weather Conditions Conducive to Infection by Cercospora beticola

To relate the dominant weather conditions thus found to reported optimum conditions conducive to CLS, we further analyzed the weather conditions based on detailed intervals of T, associated with two RH conditions (≥90% and ≥95%) and R ≥ 0.1 mm. The intervals of T were: daytime (D) T ≥ 16 °C, DT ≥ 20 °C, nighttime (N) T ≥ 10 °C, NT ≥ 15 °C, 10 °C < NT < 12 °C, and NT ≥ 18 °C. Varying weather patterns were observed according to the year, month and site (Figure 8, Appendix A). The overall dominant classes were combined NT ≥ 10 °C|R ≥ 0.1 mm|RH ≥ 90%, and combined DT ≥ 16 °C|R ≥ 0.1 mm|RH ≥ 90%. For all three study years, hourly nighttime weather conditions during the critical period of *C. beticola* infection events were dominated by temperatures varying between 10 and 18 °C, associated with RH ≥ 90% and R ≥ 0.1 mm; hourly daytime weather conditions during the critical period of *C. beticola* infection events were mostly 16 °C to 20 °C (Figure 8, Appendix A). For sites with the highest CLS incidence (Braffe 2018, Perwez 2018, Herstappe 2019 and Briffoeil 2020), dominant combined weather patterns included NT varying between 10 °C and 18 °C (few weekly total hours included conditions with NT 18 °C) and DT below 20 °C (the maximum weekly total hours including DT ≥ 20 °C never surpassed the threshold of 10 h) (Figure 8b,c, Appendix A). In 2019 and 2020, there were virtually no conditions including DT ≥ 20 °C during the months of June and July (Appendix A). Such conditions only occurred in August, with the maximum of weekly total hours peaking at 12 h in 2020 at Wagnelée (second week of August; Appendix A).

### 3.4. Performance of Weather-Based Models during the Testing Phase

Based on the dominant classes of R, RH, and daytime and nighttime T, 14 weather-based models were considered for the testing phase (Table 4). Each model was defined as a combination of defined ranges of hourly R, RH, and daytime and nighttime T over uninterrupted hours. Models with no specific rainfall condition required (absence or presence of rain) were also included to assess the importance of no continuous rain during the hours preceding a likely start of an infection event (models M13 and M14 in Table 4).

The performance of the 14 models evaluated varied. Mean values across all sites for POD and CSI ranged from 24% to 99%, and 24% and 90%, respectively; the mean FAR values ranged from 3 to 11% (Table 5). Acceptable POD and CSI (≥60%) were found for the first six models (M1 to M6; Table 5). Models M10 to M14 were those with the worst performance in predicting *C. beticola* infection events. Mean POD and CSI values for models M10 to M13 were less than 30%. Moreover, for seven out of the eight sites considered during the testing phase, the models M10 to M13 did not simulate an infection event (indicated by FAR = 0); only Rutten 2020 had a FAR value for M10 to M12 (Table 5). For model M14, all POD, FAR, and CSI values were 0 irrespective of the site, suggesting the importance of continuous rainy conditions to trigger an infection by *C. beticola* at the study sites in Belgium.

The best performing models (those resulting in relatively high POD and CSI values, and relatively low FAR values) were models M1, M2, and M3 (Table 5). For these models, POD and CSI values were mostly ≥80% for the study sites considered; FAR ranged from 0 to 20%. The three models were all characterized by a combination of continuous rainy conditions during the four hours preceding a likely start of a *C. beticola* infection event, RH > 90% during the first 4 h and RH > 60% during the following 9 h, and daytime T > 16 °C, (Table 4). Differences between models were related to the range of nighttime T: > 10 °C, >15 °C, and >18 °C, for M1, M2, and M3, respectively (Table 4). The performance of M1, M2, and M3 indicates that *C. beticola* infection events can potentially occur under nighttime temperatures close to 10 °C. Restricting RH conditions to >95% under continuous rainy conditions and relatively higher daytime T (models M10, M11, and M12) did not result in good prediction capabilities. Thus, under the environmental conditions observed during the study period, relatively hot weather conditions during the most critical infection period for *C. beticola* infection events were not conducive to development of CLS.

### 3.5. Performance of the Most Accurate Models

Models M1, M2, and M3 were considered for the validation phase since they were best performing in predicting infection events. Overall, the ability of the three models to correctly predict *C. beticola* infection events under the environmental conditions during the study period were confirmed. Models M2 and M3 had similar performance. The mean POD, CSI, and FAR values were 84% (range of 69 to 100%), 78% (range of 69 to 92%), and 6% (range of 0 to 15%), respectively (Figure 8). Extending the weather conditions to include temperatures ranging between 10 °C and 15 °C (model M1) mostly improved the POD and CSI but resulted in an increase in FAR to as much as 22% (Avernas 2020; Figure 9), indicating that some infections predicted by the model were not observed. An example of model outputs, alongside measured disease incidences, is presented (Figure 10).

At the site-year level, the analysis indicates that all three models performed well regardless of the disease incidence measured. Indeed, at Braffe and Perwez in 2018 (a year with relatively high incidence of CLS; Figure 4), the CSI and FAR were 79% and <10%, respectively, for both models M2 and M3; *C. beticola* infection events were also satisfactorily detected with a POD ≥ 79% (Figure 9). At Jandrain in 2019 (a site with relatively low incidence of CLS; Figure 5), all three models successfully detected the infection events (POD = 100%); FAR was 8% and CSI was 92% (Figure 8). At Avernas 2020, which was also a year with low disease incidence, the performance of the three models decreased slightly: POD was 85% and CSI was 73% for models M2 and M3, respectively; and FAR was 22% for model M1 (Figure 9). Such performance can be explained by the susceptibility of the cultivars sown in that year at the site, which ranged from 5 (moderately resistant) to 7 (resistant) (Table 1).

## 4. Discussion

Weather conditions conducive to CLS development in sugar beet were assessed using data collected during the 2018 to 2020 sugar beet-growing seasons at several different sites across the Belgian sugar beet belt in the Walloon Region of the country. Of the 14 weather-based models evaluated, three (models M1, M2 and M3) were outstanding and exhibited similar performance in accurately predicting *C. beticola* infection events on sugar beet at the study sites. The models’ performances indicated that a combination of continuous rainy conditions during the four hours preceding a potential *C. beticola* infection event, RH > 90% during the first four hours and RH > 60% during the following nine hours, daytime T > 16 °C and nighttime T > 10 °C, were conducive to CLS development at the study sites in Belgium. The ranges of RH favorable to *C. beticola* infection events found in this study corroborate those reported in previous studies [5,13,18]. However, the favorable daytime and nighttime temperature conditions found in this study differed to some extent from those previously reported. While daytime temperatures ranging between 23 °C and 35 °C, associated with nighttime T above 16 °C and extended periods of high RH (90 to 95%), were optimum to *C. beticola* infection events [5,13,18,44], the results from our study indicate that the ranges of favorable daytime and nighttime T can be extended to 16 °C and 10 °C, respectively, under the study conditions in Belgium. Nevertheless, such differences in favorable weather conditions conducive to CLS epidemics are expected given the differences in other environmental conditions, cultivar’s susceptibility, and quantities of inoculum in fields. Evaluating the performance of the most accurate models identified in this study under a wide range of environmental conditions and management practices would provide additional insights into the epidemiology of CLS and help improve CLS risk forecasting in temperate regions.

Our investigation highlighted the critical role of rainfall in triggering *C. beticola* infection events. Weather-based models in which continuous rainy conditions were given less importance (models M13 and M14; Table 4) were among the worst performing models (Table 5). Although there are models relying only on temperature and relative humidity to predict *C. beticola* infection and CLS progress (e.g., [28,36]), in regions with weather conditions similar to those across the study region in Belgium, attention must be paid to continuous rainy hours before a likely infection event by *C. beticola* (i.e., when optimum temperature and relative humidity conditions are met). In this study, a threshold of R ≥ 0.1 mm/h during the four preceding hours, associated with optimum RH and daytime and nighttime T, was found to provide the most favorable conditions conducive to *C. beticola* infection events. The optimum period of uninterrupted rainy hours was not investigated in this study; this should be considered in future research and would potentially help improve model performance.

The main method used to control CLS in sugar beet is regular application of fungicide. Improving fungicide-based disease management (and reducing fungicide usage) requires an approach relying on weather-based disease risk modelling rather than a growth stage-based or fixed-calendar schedules [8,36]. The results of our study are based on three sugar beet cropping seasons. With the aid of CLS monitoring data under various environmental conditions and covering a longer period of time, the selected best performing models (M1, M2, and M3) can be further validated; the modelling approach could also be fine-tuned for better model performance. Despite limitations, the model performance of M1, M2, and M3 that we demonstrated in this study can be readily embedded within a decision support tool to optimize fungicide sprays in sugar beet farms. Such simple yet effective weather-based disease prediction tools for fungal foliar diseases have been successfully implemented for managing the major fungal diseases of winter wheat in Belgium and Luxembourg [45,46], and similar models and methods can be used for sugar beet. Experimental trials have started in Belgium and France that will provide insight regarding the effectiveness of an integrated system for managing CLS in sugar beet.

In our study, CLS was first observed in sugar beet crops during the first weeks of July, most often after canopy closure. Canopy closure in sugar beet is defined as the point when leaves on 90% of beet plants in adjacent rows began to touch [8,47]. Early canopy closure in temperate climates favor higher light absorption, and potentially increased sugar yield, provided heat, and water stresses are not limiting [48,49]. A timely and reliable prediction of the time-point of canopy closure, coupled with improved prediction of *C. beticola* infection events, would ultimately benefit sugar beet farming systems.

## 5. Conclusions

Considering the damage that CLS can cause in sugar beet fields when weather conditions are favorable, it is critical to ensure that *C. beticola* infection events can be reliably predicted using weather-based disease risk models. We evaluated the performance of 14 weather-based models in predicting *C. beticola* infection events at various sites in Belgium during three consecutive sugar beet cropping seasons. Each model was defined as a combination of defined thresholds of hourly air temperature, and rainfall and relative humidity over uninterrupted hours. Six (M1 to M6) of the models evaluated predicted the infection events with acceptable accuracy, with POD, CSI, and FAR ranging, on average, from 68% to 99%, 60% to 90%, and 8% to 10%, respectively. Prediction performance for the most accurate three models (M1, M2 and M3) were validated using independent datasets, demonstrating the potential of using these weather-based models for assessing CLS risk under varying environmental conditions in Belgium. Despite limitations, the most accurate three models (M1, M2, and M3) can be readily embedded within a decision support tool to optimize fungicide sprays in sugar beet farms. Indeed, the integration of these weather-based models (forced by weather forecasts in real time) within a decision support tool can help guide judicious fungicide application to efficiently control CLS epidemics in sugar beet in Belgium and elsewhere in regions with similar environmental conditions, while minimizing unnecessary applications of fungicide.

## Figures and Tables

**Figure 1 jof-07-00777-f001:**
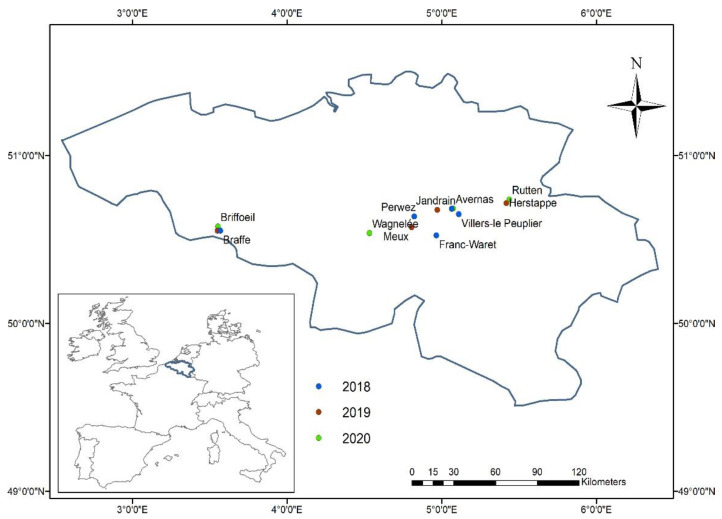
Locations of the sugar beet/Cercospora leaf spot study sites during the 2018 to 2020 cropping seasons in Belgium.

**Figure 2 jof-07-00777-f002:**
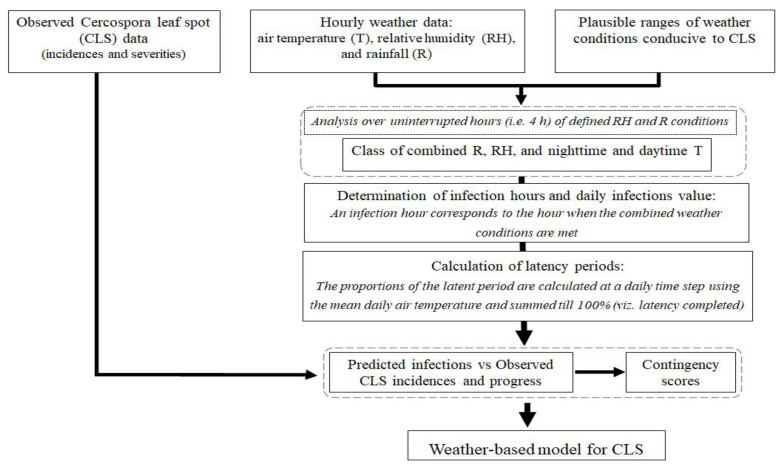
Flowchart describing the modelling approach used for predicting infection events of sugar beet by *Cercospora beticola*.

**Figure 3 jof-07-00777-f003:**
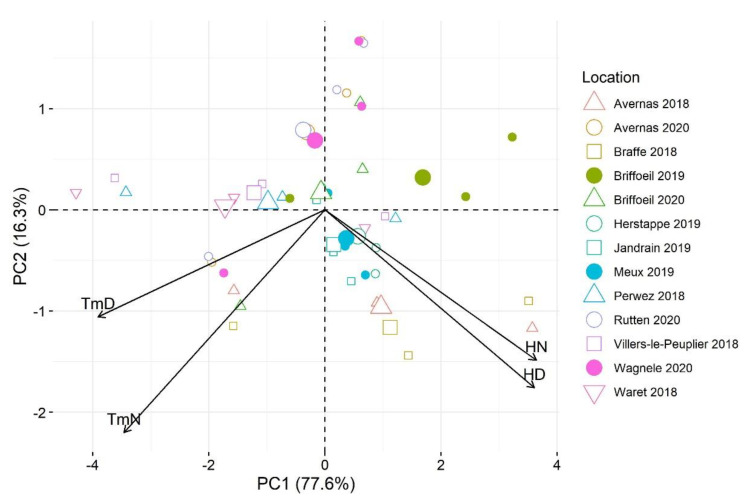
Biplot of the first two axes of the principal component analysis summarizing the relationships between climate variables and sugar beet/Cercospora leaf spot experimental sites during the June-August period of 2018–2020. TmD: mean temperature during daytime, TmN: mean temperature during nighttime; HD: mean relative humidity during daytime; and HN: mean relative humidity during nighttime. Daytime and nighttime were defined as the periods between 7.00 a.m. to 8.59 p.m. and between 9.00 p.m. to 6.59 a.m. the following day, respectively.

**Figure 4 jof-07-00777-f004:**
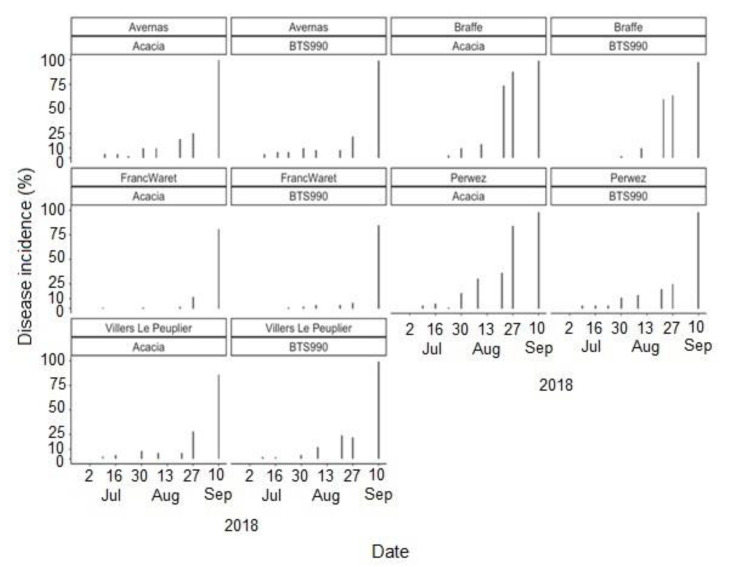
Observed incidence of Cercospora leaf spot at Avernas, Brafffe, Franc-Waret, Perwez, and Villers-le-Peuplier, Belgium, during the 2018 cropping season. Two different cultivars were monitored at each site.

**Figure 5 jof-07-00777-f005:**
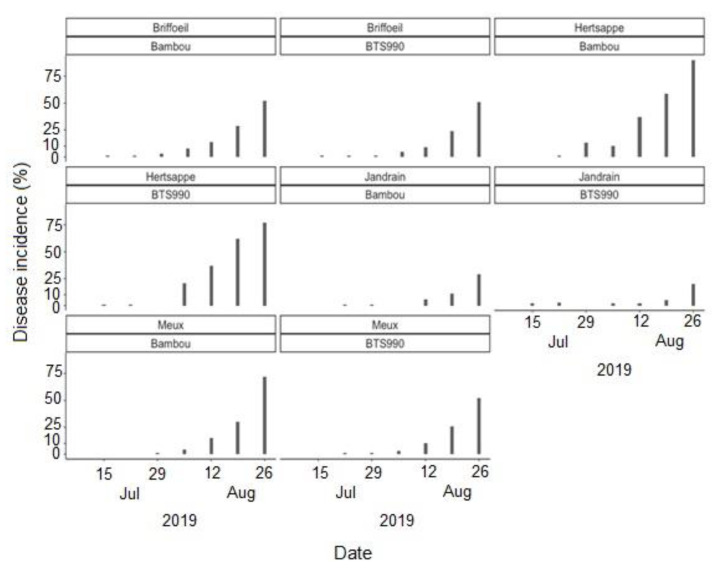
Observed incidence of Cercospora leaf spot at Briffoeil, Hertsappe, Jandrain, and Meux, Belgium, during the 2019 cropping season. Two different cultivars were monitored at each site.

**Figure 6 jof-07-00777-f006:**
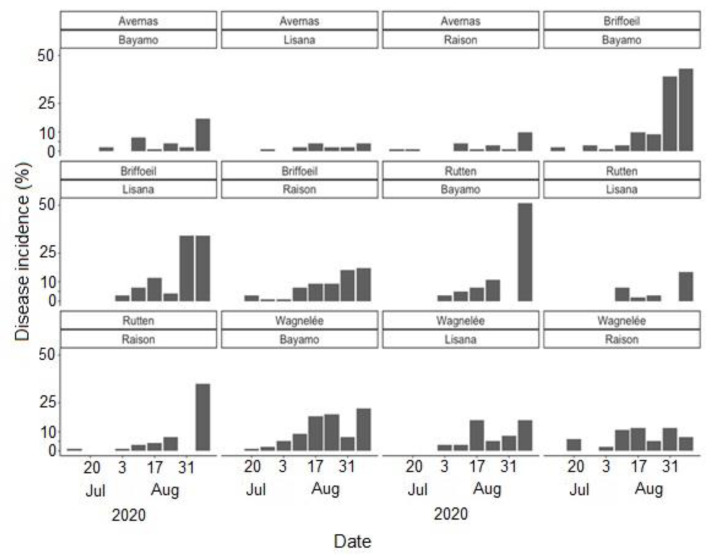
Observed incidence of Cercospora leaf spot at Avernas, Briffoeil, Rutten, and Wagnelée, Belgium, during the 2020 cropping season. Three different cultivars were monitored at each site.

**Figure 7 jof-07-00777-f007:**
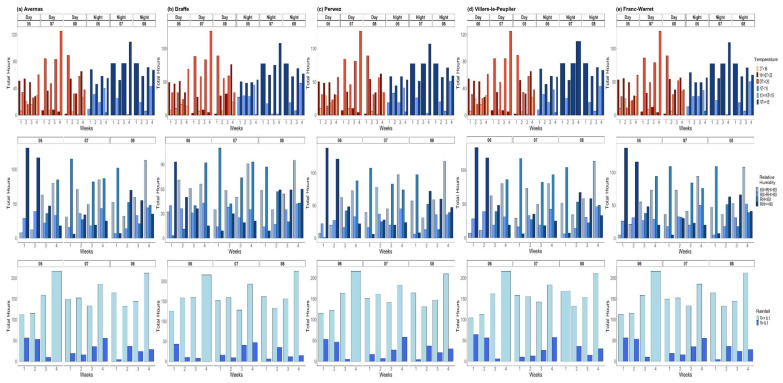
Distribution of weekly total hours for defined ranges of temperature (°C), relative humidity (%), and rainfall (mm) at Avernas (**a**), Braffe (**b**), Perwez (**c**), Villers-le-Peuplier (**d**), and Franc-Warret (**e**), Belgium, during the months of June (06), July (07), and August (08) 2018. DT and NT indicate daytime (7.00 a.m. to 8.59 p.m.) and nighttime (9.00 p.m. to 6.59 a.m. the following day) temperatures, respectively. The intervals of weather variables are provided in Table 2. (Note differences of scales on the y-axis.).

**Figure 8 jof-07-00777-f008:**
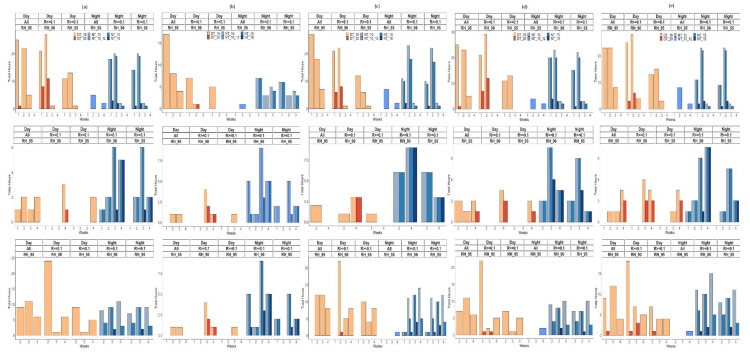
Weekly total hours of weather conditions during the months of June (top) to August (bottom) 2018 at Avernas (**a**), Braffe (**b**), Perwez (**c**), Villiers-le-Peuplier (**d**), and Franc-Waret (**e**). Weather conditions are presented as the defined combinations of rainfall (R), relative humidity (RH), and temperature (T) being met simultaneously. RH_95: RH ≥ 95%; RH_90: RH ≥ 90%; DT_16: daytime T ≥ 16 °C; DT_20: daytime T ≥ 20 °C; NT_10: nighttime T ≥ 10 °C; NT_15: nighttime T ≥ 15 °C; NT_10_12: nighttime T between 10 °C and 12 °C; NT_18: nighttime T ≥ 18 °C. Daytime: 7.00 a.m. to 8.59 p.m.; nighttime: 9.00 p.m. to 6.59 a.m. the following day. (Note differences of scales on the y-axis.).

**Figure 9 jof-07-00777-f009:**
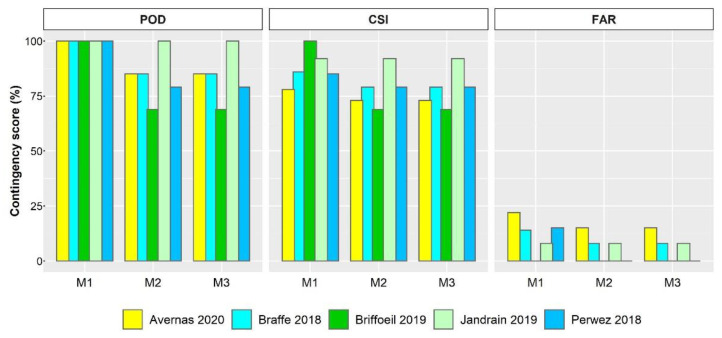
Statistical contingency scores for the three best performing weather-based models during the validation phase identified as M1, M2, M3: models #1, #2, and #3, respectively (see Table 4). POD: probability of Cercospora leaf spot detection (perfect score = 100%); CSI: critical success index (perfect score = 100%); FAR: false alarm ratio (perfect score = 0%).

**Figure 10 jof-07-00777-f010:**
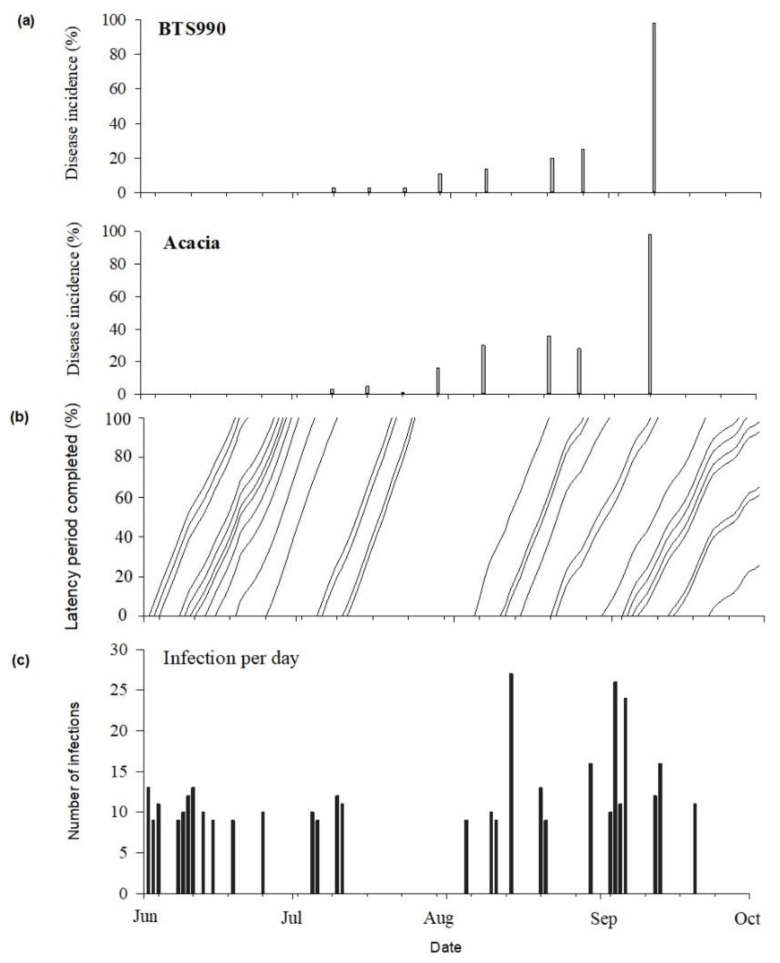
Measured incidence of Cercospora leaf spot (**a**), simulated latency period (**b**), and simulated *Cercospora beticola* infection events on sugar beet (**c**) at Perwez during the 2018 cropping season. Infection periods were simulated using the weather-based model #1 (see Table 4). The cultivars BTS990 and Acacia have a susceptibility to CLS of 6 (moderately resistant) and 4 (sensitive), respectively [30].

**Table 1 jof-07-00777-t001:** Agronomic information for the experimental fields of sugar beet used for assessing Cercospora leaf spot in Belgium during the 2018 to 2020 cropping seasons.

Year	Site	Cultivar	Sowing Date	Disease Susceptibility ^a^
2018	Avernas	BTS990	7 April 2018	6
		Acacia		4
	Braffe	BTS990	18 April 2018	6
		Acacia		4
	Franc-Waret	BTS990	16 April 2018	6
		Acacia		4
	Villers-le-Peuplier	BTS990	6 April 2018	6
		Acacia		4
	Perwez	BTS990	14 April 2018	6
		Acacia		4
2019	Briffoeil	BTS990	6 April 2019	6
		Bambou		5
		Lisanna		7
	Herstappe	BTS990	29 March 2019	6
		Bambou		5
		Lisanna		7
	Jandrain	BTS990	6 April 2019	6
		Bambou		5
		Lisanna		7
	Meux	BTS990	31 March 2019	6
		Bambou		5
		Lisanna		7
2020	Avernas	Bayamo	31 March 2020	5
		Lisanna		7
		Raison		6
	Briffoeil	Bayamo	31 March 2020	5
		Lisanna		7
		Raison		6
	Rutten	Bayamo	4 April 2020	5
		Lisanna		7
		Raison		6
	Wagnelée	Bayamo	1 April 2020	5
		Lisanna		7
		Raison		6

^a^: Cercospora leaf spot susceptibility: 1 (highly susceptible) to 9 (very resistant) [30].

**Table 2 jof-07-00777-t002:** Intervals of rainfall (R), relative humidity (RH) and air temperature (T) considered with hourly data in the frequency analysis. A class of weather variables is a combination of defined intervals of R, RH and daytime (7.00 a.m. to 8.59 p.m.) and nighttime (9.00 p.m. to 6.59 a.m. the following day) T. All times were local (GMT + 2 in summer).

Variable	Intervals			
Rainfall (mm)	R < 0.1	R ≥ 0.1		
Relative humidity (%)	RH < 60	60 ≤ RH < 80	80 ≤ RH < 90	RH ≥ 90
Daytime Temperature (°C)	T < 16	16 ≤ T < 20	T ≥ 20	
Nighttime Temperature (°C)	T < 10	10 ≤ T < 15	T ≥ 15	

**Table 3 jof-07-00777-t003:** List of sugar beet/Cercospora leaf spot experimental sites selected for model calibration and validation. Sites were randomly selected based on the principal component analysis.

Testing	Validation
Avernas 2018	Braffe 2018
Franc-Waret 2018	Perwez 2018
Villers-le-Peuplier 2018	Briffoeil 2019
Herstappe 2019	Jandrain 2019
Meux 2019	Avernas 2020
Briffoeil 2020	
Wagnelée 2020	
Rutten 2020	

**Table 4 jof-07-00777-t004:** List of the models evaluated during the testing step. Each model was defined as a combination of defined thresholds of hourly air temperature, and rainfall and relative humidity over uninterrupted hours.

Model	Hourly Rainfall (R)	Hourly Relative Humidity (RH)	Hourly Temperature (T; °C)	
			Nighttime	Daytime
M1	R ≥ 0.1 mm during the 4 preceding hours	RH > 90% during the first 4 h, then >60% during the following 9 h	T > 10	T > 16
M2			T > 15	T > 16
M3			T > 18	T > 16
M4	R ≥ 0.1 mm during the 4 preceding hours	RH > 95% during the first 4 h, then >60% during the following 9 h	T > 10	T > 16
M5			T > 15	T > 16
M6			T > 18	T > 16
M7	R ≥ 0.1 mm during the 4 preceding hours	RH > 90% during the first 4 h, then >60% during the following 9 h	T > 10	T > 20
M8			T > 15	T > 20
M9			T > 18	T > 20
M10	R ≥ 0.1 mm during the 4 preceding hours	RH > 95% during the first 4 h, then >60% during the following 9 h	T > 10	T > 20
M11			T > 15	T > 20
M12			T > 18	T > 20
M13	No specific rainfall condition required	RH > 95% during the first 4 h, then >60% during the following 9 h	10< T < 12	T > 16
M14			10< T < 12	T > 20

**Table 5 jof-07-00777-t005:** Statistical contingency scores for the three best performing weather-based models during the testing phase. M1–13: model #1 to model #13, respectively. POD: probability of Cercospora leaf spot detection (best score = 100%); FAR: false alarm ratio (best score = 0%); CSI: critical success index (best score = 100%). The 3 best performing models are underlined (M1, M2 and M3). All the POD, FAR, and CSI values for model M14 were 0 (no simulation achieved based on the combined weather conditions defined for that model); the statistics were therefore not presented.

	M1	M2	M3	M4	M5	M6	M7	M8	M9	M10	M11	M12	M13
	Probability of detection (POD)												
Avernas 2018	100	73	80	75	75	75	33	33	33	22	20	20	20
Briffoeil 2020	100	94	95	88	88	88	72	73	73	36	36	36	38
Herstappe 2019	100	92	92	100	100	100	56	56	56	25	22	22	22
Meux 2019	100	83	83	50	50	50	44	44	44	29	29	29	25
Wagnéllie 2020	100	95	95	54	54	54	69	69	69	36	36	36	22
Waret 2018	91	82	80	90	73	73	43	38	38	29	29	29	43
Villers-le-Peuplier 2018	100	100	100	75	75	75	46	46	46	22	22	22	22
Rutter 2020	100	100	100	29	29	29	91	91	91	14	14	14	0
Mean—All sites	99	90	91	70	68	68	57	56	56	27	26	26	24
	False alarm ratio (FAR)												
Avernas 2018	14	20	11	0	0	0	0	0	0	0	0	0	0
Briffoeil 2020	8	10	16	0	0	0	0	0	0	0	0	0	0
Herstappe 2019	14	14	8	0	0	0	17	17	17	0	0	0	0
Meux 2019	0	0	0	0	0	0	0	0	0	0	0	0	0
Wagnéllie 2020	10	10	10	13	13	13	0	0	0	0	0	0	0
Waret 2018	0	11	11	0	0	11	0	0	0	0	0	0	0
Villers-le-Peuplier 2018	20	20	20	0	0	0	0	0	0	0	0	0	0
Rutter 2020	6	6	6	50	50	50	10	10	10	50	50	50	0
Mean—All sites	9	11	10	8	8	9	3	3	3	6	6	6	0
	Critical success index (CSI)												
Avernas 2018	86	62	73	75	75	75	33	33	33	22	20	20	20
Briffoeil 2020	92	85	81	88	88	88	72	73	73	36	36	36	38
Herstappe 2019	86	80	86	60	50	50	50	50	50	25	22	22	22
Meux 2019	100	83	83	50	50	50	44	44	44	29	29	29	25
Wagnéllie 2020	90	86	86	50	50	50	69	69	69	36	36	36	22
Waret 2018	91	75	73	90	73	67	43	38	38	29	29	29	43
Villers-le-Peuplier 2018	80	80	80	75	75	75	46	46	46	22	22	22	22
Rutter 2020	94	94	94	22	22	22	83	83	83	13	13	13	0
Mean—All sites	90	81	82	64	60	60	55	55	55	27	26	26	24

## Data Availability

The geographical boundaries of the study fields are not publicly available due to privacy protection. The datasets supporting this article are available from the corresponding author upon reasonable request.

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
