# Peer review of "Weather-Based Predictive Modeling of Cercospora beticola Infection Events in Sugar Beet in Belgium"

_jof, 2021, doi:10.3390/jof7090777_

Round 1
Reviewer 1 Report
Remarks
L 346: The description for table 5 indicates the presentation of 14 models, but this table covers only 13 models.
In Table 4-6: It is worthy to unify the way of data presentation.
To the summary should be added the recommendation about the fungicide treatment timing/date in regarding the described models.
Author Response
(The number of lines indicated in the responses refer to the revised manuscript with track changes.)
Reviewer 1 |
|
Comment |
Response |
Remarks
L 346: The description for table 5 indicates the presentation of 14 models, but this table covers only 13 models.
|
We appreciate the reviewer’s comments. The sentence in L346 was revised to indicate the 13 models presented. See L346 of the revised version of the manuscript.
Although 14 models were evaluated, the statistics for model #14 were not presented in Table 5 because all the POD, FAR and CSI values for this model were 0; no simulation was achieved based on the combined weather conditions defined for that model.
We have rewritten the sentence in the revised manuscript (see L349-350).
|
In Table 4-6: It is worthy to unify the way of data presentation.
|
We have changed the format of Table 4, which presents the models (i.e. combinations of weather conditions to be met), for clarity.
In Table 5 we presented the performance statistics of the models evaluated during the testing phase for each of the sites selected for this purpose. As indicated in the previous response, the statistics for model #14 were not shown in the table because all values were equal 0 (no simulation achieved based on the combined weather conditions defined for that model). This aspect was emphasized in the table caption.
|
To the summary should be added the recommendation about the fungicide treatment timing/date in regarding the described models.
|
A sentence was inserted into Section 5 to address the comment. Se L458-460 of the revised manuscript. |

Reviewer 2 Report
Congratulations to the good choice of research concept and to your work on this well developed MS.
Please find my few comments on your MS:
- Fig. 3. Its quality is not acceptable and its symbols are not unambiguous (e.g. simple triangle, simple circle).
- Table 4. There is need for an improvement on its face.
- Fig. 10/b. seems to be without dimensions.
- Please use Italic for latin names in References: 1, 3, 6, 7, 12, 19, 25, 28.
- Small typographical or grammatical errors: r. 498, r. 535 (eds.?), r. 546.
Author Response
(The number of lines indicated in the responses refer to the revised manuscript with track changes.)
Reviewer 2 |
|
Comment |
Response |
Congratulations to the good choice of research concept and to your work on this well-developed MS.
Please find my few comments on your MS:
|
We appreciate the reviewer’s comments. We have addressed all comments and revised the manuscript accordingly. |
Fig. 3. Its quality is not acceptable and its symbols are not unambiguous (e.g. simple triangle, simple circle).
|
Figure 3 has been redrawn for clarity. See new Figure 3 in the revised manuscript. |
Table 4. There is need for an improvement on its face.
|
We have changed the format of Table 4 for clarity (i.e. duplication of rainfall conditions for models M#1 to M#12; insertion of dotted lines to separate models with similar rainfall and relative humidity conditions).
|
Fig. 10/b. seems to be without dimensions.
|
Thank you for pointing this out. The y-axis has been added to the figure (see new Figure 10 of the revised manuscript). This y-axis refers to the latency achieved and ranges from 0 to 100%.
|
Please use Italic for Latin names in References: 1, 3, 6, 7, 12, 19, 25, 28.
|
Done. |
Small typographical or grammatical errors: r. 498, r. 535 (eds.?), r. 546.
|
All the relevant changes have been made in the revised manuscript. |

Reviewer 3 Report
The paper is well documented and presented. The studies are of great interest and benefit to farmers
Author Response
Thank you for the comments. Very much appreciated.